# Synthesis of 3-(2-Alkylthio-4-chloro-5-methylbenzenesulfonyl)-2-(1-phenyl-3-arylprop-2-enylideneamino)guanidine Derivatives with Pro-Apoptotic Activity against Cancer Cells

**DOI:** 10.3390/ijms24054436

**Published:** 2023-02-23

**Authors:** Aneta Pogorzelska, Jarosław Sławiński, Anna Kawiak, Grzegorz Stasiłojć, Jarosław Chojnacki

**Affiliations:** 1Department of Organic Chemistry, Medical University of Gdańsk, Al. Gen. J. Hallera 107, 80-416 Gdańsk, Poland; 2Department of Biotechnology, Intercollegiate Faculty of Biotechnology, University of Gdańsk and Medical University of Gdańsk, Abrahama 58, 80-307 Gdańsk, Poland; 3Department of Cell Biology and Immunology, Intercollegiate Faculty of Biotechnology of UG and MUG, Medical University of Gdańsk, Dębinki 1, 80-211 Gdańsk, Poland; 4Department of Inorganic Chemistry, Gdańsk University of Technology, Narutowicza 11/12, 80-233 Gdańsk, Poland

**Keywords:** benzenesulfonamides, benzenesulfonylguanidines, amidrazones, anticancer activity, cytotoxicity, cell cycle flow cytometry analysis, apoptosis, membrane potential

## Abstract

The untypical course of reaction between chalcones and benzenesulfonylaminoguanidines led to the new 3-(2-alkylthio-4-chloro-5-methylbenzenesulfonyl)-2-(1-phenyl-3-arylprop-2-enylideneamino)guanidine derivatives **8**–**33**. The new compounds were tested in vitro for their impact on the growth of breast cancer cells MCF-7, cervical cancer cells HeLa and colon cancer cells HCT-116 by MTT assay. The results revealed that the activity of derivatives is strongly related to the presence of hydroxy group in the benzene ring at the 3-arylpropylidene fragment. The most cytotoxic compounds **20** and **24** displayed mean IC_50_ values of 12.8 and 12.7 μM, respectively, against three tested cell lines and were almost 3- and 4-fold more active toward MCF-7 and HCT-116 when compared with non-malignant HaCaT cells. Furthermore, compound **24** induced apoptosis in cancer cells and caused a decrease of mitochondrial membrane potential as well as an increase of cells in sub-G1 phase in contrast to its inactive analog **31**. The strongest activity against the most sensitive HCT-116 cell line was found for compound **30** (IC_50_ = 8 μM), which was 11-fold more effective in the growth inhibition of HCT-116 cells than those of HaCaT cells. Based on this fact, the new derivatives may be promising leading structures for the search for agents for the treatment of colon cancer.

## 1. Introduction

Cancer is becoming a global challenge. In 2020, the number of new cases worldwide has exceeded 19 million, with the number of deaths reaching nearly 10 million. These records are expected to rise, and by 2040, the number of new cases will reach 30.2 million and the number of deaths will increase to 16.3 million [1]. Although cancer incidence is mainly related to environmental factors, mortality is primarily due to late diagnosis and a lack of effective treatments. Among the currently available methods to treat cancer, the most common is the use of small molecule drugs, which have many advantages compared to biologics, such as the pharmacokinetic properties, patient compliance, easier administration, costs, and drug storage [2].

The structures of all new small-molecule drugs approved from 2015 to 2020 for cancer treatment are characterized by polynitrogen motifs [3]. A large library of cytotoxic compounds against various types of cancer consist of amidrazone/guanidine groups as a polynitrogen motif. These structures have been found in natural anticancer alkaloids, such as crambescidines or uropocidin C [4,5]. A similar motif is present in the structure of a lot of synthethic low molecular weight compounds with cytotoxic activity depicted in Figure 1 [6,7,8,9,10,11,12,13,14]. Our previous research proved that the polynitrogen fragment is an important element of benzenesulfonamides with antiproliferative activity [15,16,17,18,19,20]. The amidrazone scaffold can also be found among benzensesulfonylguanidine derivatives synthesized by our team. Previously, we reported significant cytotoxic activity against human cancer cell lines or amidrazone derivatives modified by the alkynyl scaffold [17]. These results prompted us to the design new guanidines with potential anticancer activity that will be discussed below.

## 2. Results and Discussion

### 2.1. Chemistry

The synthesis of new 3-(2-alkylthio-4-chloro-5-methylbenzenesulfonyl)-2-(1-phenyl-3-arylprop-2-enylideneamino)guanidine **8**–**33** has been presented at Figure 1. The appropriate substrates, 1-amino-2-(2-alkylthio-4-chloro-5-methylbenzenesulfonyl)guanidines (**1**–**7**), were obtained as previously described [6,21,22,23,24]. As shown, the final products **8**–**33** were obtained via reaction of guanidines **1**–**7** with a suitable chalcone derivative.

The structures of final compounds **8**–**33** were confirmed with spectroscopic methods IR, ^1^H, and ^13^C NMR, elemental analyses, and mass spectrometry.

IR spectra of compounds **8**–**33** showed absorption bands derived from NH bonds in the ranges 3493–3183 cm^−1^ and 1608–1653 cm^−1^. The bands at range 1321–1342 cm^−1^ and 1125–1176 cm^−1^ were due to an SO_2_ group.

The ^1^H NMR spectra of the series of 3-(2-alkylthio-4-chloro-5-methylbenzenesulfonyl)-2-(1-phenyl-3-arylprop-2-enylideneamino)guanidine **8**–**33** showed singlets at the range of 2.26–2.30 ppm for three protons of a methyl group. A singlet corresponding to two protons in the range of 4.21−4.33 ppm was due to a thiomethyl group. The characteristic multiplet observed in the spectra of all compounds **8**–**33** is a doublet integrated for one proton in the range of 6.20–6.60 ppm with the coupling constant J_H2-H3_ = 16.1–16.6 Hz that was due to a H-2 proton (CH=C) of the double bond. The H-3 proton (C=CH) of this moiety was found at around 6.90–7.40 ppm either in multiplets or as a doublet with the corresponding coupling constant. Protons H-3 and H-6 from benzenesulfonyl scaffold appeared at the range of 7.40–7.48 ppm and 7.49–7.58 ppm, respectively. Other characteristic features of the ^1^H NMR spectra of novel derivatives **8**–**33** are singlets with integration for 1H at chemical shift ranges 7.52–7.70 ppm, 7.66–7.82 ppm, and 9.79–10.07 ppm, which were due to NH protons.

The final confirmation of the product structure was provided by the crystallographic analysis. The data clearly indicated that the obtained products are unusual for the standard reaction between the NH-NH fragment and the chalcone derivatives. According to the literature, hydrazines react with chalcones to form pyrazoline ring [10,11,12,13,14,15,16,17,18,19,20]. In the case of benzenesulfonylaminoguanidines, the obtained aminoazadiene fragment does not undergo spontaneous cyclization in the standard conditions, regardless the reaction time. This was proven by X-ray diffraction studies for exampling compounds **24** and **31**.

Compound **24** forms brownish, transparent crystals satisfying symmetry of the monoclinic system, the space group $P2_{1}/c$ (no. 14). The asymmetric unit contains one molecule of sulfonamide and one molecule of acetone linked by hydrogen bonding. Molecular structure is shown in Figure 2. The main molecule contains a 2-chlorofenylmetyltio group disordered over two positions with site occupation factors of 0.646(6)/0.354(6). Most of the bond lengths and angles are in the expected ranges. Crystal data, data collection, and structure refinement details are summarized in Appendix A. The sulfonamide is deprotonated at N1 and the proton is transferred to N2 making N1 negatively charged and N2 positively charged (the zwitterionic form). This enables formation of strong charge-assisted hydrogen bonding in dimers present in the solid state (Figure 3). Additionally, the internal NH…O cyclic hydrogen bond S(6) stabilizes the conformation. The two pending hydroxyl groups bond two acetone molecules on both sides of the associate. These entities are then packed in the solid state hold by weaker Wan der Waals forces.

Compound **31** forms light-brown, transparent crystals satisfying the symmetry of the triclinic system, the space group $P\bar{1}$ (no. 2). The asymmetric unit contains one molecule of the protonated sulfonamide (cation) and one tosyl anion; no solvent is present this time. Molecular structure is shown in Figure 4. The tosyl sulfonic group SO_3_ is disordered over two positions with site occupation factors of 0.64(3)/0.36(3). Crystal data, data collection, and structure refinement details are summarized in Appendix A. The sulfonamide is protonated at N1 and two hydrogen atoms are attached to N2. Based on bond lengths, we can assume the single-bond character of C1-N1 and C1-N2 and the double-bond character of C1=N3, making N3 formally positively charged. Then we find the formation of strong, cyclic, charge-assisted hydrogen bonding between two cations and two anions present in the solid state (Figure 5). The hydrogen bond motif can be topologically classified as $R_{4}^{4}\left(16\right)$ or as NH…N (using N1 and N2 as hydrogen donors and O5 and O6 from tosyl anion as acceptors) located at the inversion center (drawn as the orange ball). Additionally, the internal NH…O cyclic hydrogen bond S(6) stabilizes the cation conformation similarly as found in **24**. These supramolecules are then packed in the solid state by other Wan der Waals forces.

### 2.2. Cytotoxic Activity

The cytotoxic activity of the compounds **8**–**33** has been evaluated in MTT assays against three human cancer cell lines MCF-7 (breast cancer), HeLa (cervical cancer), and HCT-116 (colon cancer) and it was expressed as IC_50_ values in μM (Table 1).

The results indicated that cytotoxic activity is observed only for compounds with a hydroxyl group as an R^2^ substituent. Compounds **20** and **24** displayed the best cytotoxic properties with mean IC_50_ values of 12.8 and 12.7 μM, respectively, against three tested cell lines. Derivatives **10**, **14**, **15**, **20**, **24**, **26**, and **30** inhibited the growth of HCT-116 cells with IC_50_ values in the range of 8–10 μM. Although the highest activity of the compounds with hydroxyl groups as the R^2^ residue was against colon cancer cells, the inhibition of the growth of breast cancer cells was also remarkable. All compounds with R^2^ = OH showed IC_50_ in the range of 12–20 μM against the MCF-7 cell line, and derivatives **14**, **15**, **24**, and **26** were the most promising (IC_50_ = 12 μM for **24** and 13 μM for **14**, **15**, and **26**).

The important feature of new cytotoxic agents is their selective activity against cancer cells with a weak impact on the physiology of healthy cells. Determination of the selectivity of compounds **14**–**19** and **24**–**29** was done by MTT assay with the non-cancerous HaCaT cell line (Table 2). The obtained data indicate that in the majority, compounds show significantly higher activity against cancer cells, especially HCT-116, in comparison with HaCaT cells. What is important is that the cytotoxicity against non-cancerous cells of synthesized derivatives was also significantly lower than the reference drug, cisplatin (Table 1). Compounds **20** and **24** with the best cytotoxic profile displayed also remarkable selectivity toward cancer cells in comparison with non-cancerous HaCaT cells, with a selectivity ratio nearly three and four times higher for MCF-7 and HCT-116, respectively. A similar effect was observed also for compound **26**. Importantly, the mean IC_50_ value for compound **20** and **24** against cancer cells (12.8 μM and 12.7 μM) was also significantly lower than the IC_50_ for HaCaT (31 μM and 33 μM, respectively; selectivity ratio 2.42 and 2.6, respectively). Importantly, compound **30** displayed the highest selectivity inhibiting the growth of MCF-7 and HCT-116 cells almost 5 and 11 times better, respectively, than the non-malignant cells. Importantly, the selectivity of derivative **30** was significantly better when compared to cisplatin (Table 2, selectivity index 2.56 for MCF-7 and 2.02 for HCT-116). Due to the excellent selectivity of compound **30** towards HCT-116 cells and at the same time high cytotoxicity towards this cell line (IC_50_ = 8 μM); derivative **30** may be a good hit compound for the search for chemotherapeutic agents for the treatment of colon cancer.

### 2.3. Apoptotic Activity

Agents that induce apoptosis are believed to be the most effective non-surgical treatment of cancer. Therefore, the biochemical markers, such as DNA fragmentation, loss of mitochondrial membrane potential (Δψm), and phosphatidylserine translocation, were investigated. The changes in cell morphology were also examined. The experiments were done for compound **24** with the strongest cytotoxicity as well as the inactive derivative **31**. The studies of compound **24** were performed in a concentration-dependent manner using compound concentrations of 10 μM and 25 μM. The inactive analog **31** was tested only at a higher concentration of 25 μM. The charts showing the results of the cytometric analysis are presented in Figure 6.

#### 2.3.1. Cell Morphology

The evaluation of changes in cell morphology has been performed after incubation with compounds **24** and **31** for 72 h using light microscopy. The apoptotic-like changes, such as shrinkage of the cells or detachment from the surface, were observed in the morphology of tested cells treated with both concentrations of **24** in contrast to the inactive analog **31** (Figure 7).

#### 2.3.2. Cell Cycle Analysis

The cell cycle distribution was measured by flow cytometry analysis after incubation of MCF-7, HeLa, and HCT-116 with compounds **24** and **31** for 72 h.

The results shown in Figure 8 indicate the significant increase in the cell distribution in the sub-G1 phase in a dose-dependent manner in the **24**-treated HCT-116 and HeLa cells (HCT-116 from 1.15% in control to 65.31%; HeLa from 5.08% in control to 18.07%). Although compound **24** affects the cell cycle of HCT-116 cells already at a concentration of 10 μM (46.69% of cells in sub-G1 phase in contrast to 1.15% in control), a slightly weaker effect is observed for HeLa cells for which a sub-G1 fraction was observed after treatment with 25 µM of **24**. In the case of MCF-7 cells treated with **24**, no significant effect on the cell cycle distribution was observed, regardless of the amidine concentration. Derivative **31**, in turn, does not substantially affect the cell cycle progression of any of the tested cell lines.

#### 2.3.3. Mitochondrial Membrane Potential (ΔΨm) Analysis

A decrease in mitochondrial potential is one of the earliest hallmarks of apoptosis. A common method for indication of cells with high and low Δψm is flow cytometry combined with specific fluorescent probes, such as JC-1 (5,5′,6,6′-tetrachloro-1,1′,3,3′-tetraethylbenzimidazolylcarbocyanine iodide). JC-1 accumulates in mitochondria and forms complexes (J aggregates) with red fluorescence in healthy cells with a normal ΔΨ_m_, whereas in apoptotic cells with a low ΔΨ_m_ exists as a monomer with green fluorescence. 

Flow cytometry analysis of MCF-7, HeLa, and HCT-116 cells treated with **24** showed a remarkable decrease in ΔΨ_m_, even at a low concentration of compound, as shown in Figure 9. By contrast, exposure of cells to inactive derivative **31** did not result in a loss of ΔΨ_m_.

#### 2.3.4. Translocation of Phosphatidylserine to Outer Leaflet of Cell Membrane

One of the early indicators of apoptosis is the exposure of phosphatidylserine residues on the cell’s surface. Annexin V has a high affinity to phosphatidylserine and, as a conjugate to fluorescein isothiocyante (FITC), is widely used for detection of early apoptotic cells using flow cytometry. Detection of late-apoptotic or necrotic cells is achieved by using propidium iodide (PI) stain. Based on the obtained fluorescence, four subpopulations were found: PI-low/FITC-low (live cells), PI-high/FITC-low (necrotic cells), PI-low/FITC-high (early apoptotic cells), and PI-high/FITC-high (late apoptotic cells).

As shown in Figure 10, an increased population of apoptotic cells appeared in all tested cell lines treated with compound **24** in a dose-dependent manner. Induction of apoptosis was noticed for MCF-7 cells and HCT-116 cells exposed to **24** already at a concentration of 10 μM, and this effect was improved when exposed to a concentration of 25 μM. A significantly increased level of late apoptotic cells in the HeLa cell line was observed only after treatment with 25 μM of **24**. Importantly, incubation of cells with inactive amidine **31** did not cause statistically significant differences compared to the control. 

## 3. Materials and Methods

### 3.1. Synthesis

The procedures for the preparation and spectral characteristic of compounds **8**–**33** are provided in the Appendix A.

### 3.2. Crystallographic Details

Diffraction intensity data for **24** and **31** were collected on an IPDS 2T dual beam diffractometer (STOE & Cie GmbH, Darmstadt, Germany) at 120.0(2) K with MoKa radiation of a microfocus X-ray source (GeniX 3D Mo High Flux, Xenocs, Sassenage, 50 kV, 0.6 mA, and λ = 0.71069 Å). Investigated crystals were thermostated under a nitrogen stream at 120 K using the CryoStream-800 device (Oxford CryoSystem, Long Hanborough, Oxford, UK) during the entire experiment.

Data collection and data reduction were controlled by using the X-Area 1.75 program (STOE, 2015, Darmstadt, Germany). Numerical absorption correction was not performed due to low absorption. The structure was solved using intrinsic phasing implemented in SHELXT and refined anisotropically using the program packages Olex2 [24] and SHELX-2015 [25,26]. Positions of the C–H hydrogen atoms were calculated geometrically taking into account isotropic temperature factors. All H-atoms were refined as riding on their parent atoms with the usual restraints.

Structure **24** was refined with usual procedures; structure **31** was refined as a two-component twin, with the fraction of domains equal to 0.587(6) and 0.413(6).

Crystallographic data for all structures reported in this paper have been deposited with the Cambridge Crystallographic Data Centre as supplementary publication Nos. CCDC 2213778-2213779. The data can be obtained free of charge from The Cambridge Crystallographic Data Centre via www.ccdc.cam.ac.uk/structures (accessed on 19 Octobert 2022).

### 3.3. Cell Culture and Cell Viability Assay

All chemicals, if not stated otherwise, were obtained from Sigma-Aldrich (St. Louis, MO, USA). The MCF-7 cell line was purchased from Cell Lines Services (Eppelheim, Germany), the HeLa and HCT-116 cell lines were obtained from the Department of Microbiology, Tumor and Cell Biology, Karolinska Institute (Stockholm, Sweden). Cells were cultured in Dulbecco’s modified Eagle’s medium (DMEM) supplemented with 10% fetal bovine serum, 2 mM glutamine, 100 units/mL penicillin, and 100 μg/mL streptomycin. Cultures were maintained in a humidified atmosphere with 5% CO_2_ at 37 °C in an incubator (HeraCell, Heraeus, Langenselbold, Germany).

Cell viability was determined using the MTT (3-(4,5-dimethylthiazol-2-yl)-2,5-diphenyl-tetrazoliumbromide) assay. Stock solutions of the studied compounds were prepared in 100% DMSO. Working solutions were prepared by diluting the stock solutions with DMEM medium, the final concentration of DMSO did not exceed 0.5% in the treated samples. Cells were seeded in 96-well plates at a density of 5 × 10^3^ cells/well and treated for 72 h with the examined compounds in the concentration range 1–100 μM (1, 10, 25, 50, and 100 μM). Following treatment, MTT (0.5 mg/mL) was added to the medium and cells were further incubated for 2 h at 37 °C. Cells were lysed with DMSO and the absorbance of the formazan solution was measured at 550 nm with a plate reader (1420 multilabel counter, Victor, Jügesheim, Germany). The optical density of the formazan solution was measured at 550 nm with a plate reader (1420 multilabel counter, Victor, Jügesheim, Germany). The experiment was performed in triplicate. Values are expressed as the mean ± SD of at least three independent experiments.

#### 3.3.1. Cell Morphology

The HeLa, MCF-7, and HCT-116 cells were seeded on 24-well plates (5 × 10^4^/per well) in 1 mL of medium for 24 h. After that time, cells were treated with **24** and **31** for next 72 h. Cells were treated with 10 µM, 25 µM of **24,** or 25 µM of **31**. As a control, non-treated cells were used. Morphology of cells was observed after 72 h using light microscope Olympus IX83.

#### 3.3.2. Mitochondrial Membrane Potential (Δψm) Analysis

Analyzed cells were seeded into wells of a 24-well plate at a density of 5 × 10^4^ cells in 1000 µL, incubated overnight, and then the medium was exchanged for dilutions of reagents **24** and **31**. Before the end of 72 h of incubation, MitoProbe JC-1 (25 µM) was added into each plate well. Carbonyl cyanide m-chlorophenylhydrazone CCCP (200 nM) as the mitochondrial oxidative phosphorylation uncoupler was added into the positive control 15 min before JC-1 was added. After 30 min of JC-1 staining, cells were washed with PBS and then trypsinized (Corning^®^ 25-053CI). Cells suspended in PBS were analyzed by flow cytometry at λ excitation (ex) = 488 nm and λ emission (em) = 525/570 nm (LSR II BD Biosciences, San Jose, CA, USA).

#### 3.3.3. Translocation of Phosphatidylserine to the Outer Leaflet of Cell Membrane

Analysis was performed using FITC-conjugated annexin-V (Annexin V-FITC Apoptosis Kit I, BD Biosciences, San Jose, CA, USA) according to the manufacturer’s instructions (BD Biosciences) as described previously [27]. HeLa, MCF-7, and HCT-116, after seeding to 24-well plates (5 × 10^4^/per well) and 24 h incubation, were treated for 72 h with **24** or **31** (10/25 µM or 25 µM concentration respectively). Following treatment, cells were washed in PBS, centrifuged, and then resuspended in a binding buffer. Afterward, the cells were incubated for 15 min at 37 °C with FITC-conjugated annexin-V and propidium iodide. The samples were then analyzed using a LSR II flow cytometer (BD Biosciences) using 530 ± 25 nm (Annexin V-FITC) and 575 ± 26 nm (PI).

The subpopulations were identified according to their fluorescence: PI-low/FITC-low (live cells), PI-high/FITC-low (necrotic cells), PI-low/FITC-high (early apoptotic cells), and PI-high/FITC-high (late apoptotic cells).

#### 3.3.4. Statistical Analysis

Statistical differences between control and treated cells were determined using the One-Way ANOVA test followed by Dunn’s post hoc test with Bonferroni corrected *p* values. The analyses were performed using 9 to 15 replicates run in at least three independent experiments. Statistical analysis was performed using PAST 4.0.

## 4. Conclusions

We synthesized a new series of 3-(2-alkylthio-4-chloro-5-methylbenzenesulfonyl)-2-(1-phenyl-3-arylprop-2-enylideneamino)guanidine derivatives **8**–**33,** which have been tested for their antiproliferative activity against three cancer cell lines: colon HCT-116, cervical HeLa, and breast MCF-7. The results indicated that the activity of the synthesized compounds was entirely dependent on the R^2^ substituent and occurred only when R^2^ = OH. The highest sensitivity to compounds witrh a hydroxy group in phenyl, substituted at position 4 of the amidrazone scaffold, was noticed for the colon cancer cell line HCT-116. HeLa and MCF-7 cells were slightly less sensitive. 3-{4-Chloro-2-[(4-methylphenyl)methylthio]-5-methylbenzenesulfonyl}-2-[3-(4-hydroxyphenyl)-1-phenylprop-2-enylideneamino]guanidine (**20**) and 3-{4-chloro-2-[(2-chlorophenyl)methylthio]-5-methylbenzenesulfonyl}-2-[3-(2-hydroxyphenyl)-1-phenylprop-2-enylideneamino]guanidine (**24**) were the compounds with the strongest cytotoxicity against tested cancer cell lines and good selectivity in comparison with their activity toward normal HaCaT cells. 

The cytotoxicity of novel compounds is associated with the induction of apoptosis, especially on HCT-116 and MCF-7 cells as was shown in the studies with Annexin V and analysis of the mitochondrial membrane potential after treatment of cancer cells with compound **24** and its inactive analog **31**.

Despite the high activity of compounds **20** and **24**, the structure with the best parameters against the most sensitive HCT-116 cells was 3-{4-chloro-2-[(4-chlorophenyl)methylthio]-5-methylbenzenesulfonyl}-2-[3-(4-hydroxyphenyl)-1-phenylprop-2-enylideneamino]guanidine (**30**) with the lowest IC_50_ (8 μM) and the highest selectivity when compared to non-cancerous HaCaT cells (11 times stronger growth inhibition of HCT-116 then HaCaT). Taking into account cytotoxicity and selectivity toward HCT-116 cells, compound **30** is a good leading structure for the search of new agents for the treatment of colon cancer.

## Data Availability

All data are available as Appendix A.

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
