# Peer review of "Synthesis of 3-(2-Alkylthio-4-chloro-5-methylbenzenesulfonyl)-2-(1-phenyl-3-arylprop-2-enylideneamino)guanidine Derivatives with Pro-Apoptotic Activity against Cancer Cells"

_ijms, 2023, doi:10.3390/ijms24054436_

Round 1

Reviewer 1 Report

The manuscript entitled “Synthesis and Molecular Structure of 3-(2-alkylthio-4-chloro-5-methylbenzenesulfonyl)-2-(1-phenyl-3-arylprop-2-enylideneamino)guanidine Derivatives - New Pro-apoptotic Agents with Cytotoxic Activity Against Cancer Cells” deals with the synthesis, characterization and anticancer activities (against colon cancer cells HCT-116, cervical cancer cells HeLa, breast cancer cells MCF-7) of the new 3-(2-alkylthio-4-chloro-5-methylbenzenesulfonyl)-2-(1-phenyl-3-arylprop-2-enylideneamino)guanidine derivatives.

The characterization results indicate that the compounds were successfully synthesized and characterized. The compound 24 and 31 were also characterized in detail by using X-ray diffraction analysis and the structures were elucidated clearly. As a structure-activity relationship, they concluded that biological activity depends strongly on the presence of hydroxy group in benzene ring in the 3-arylpropylidene moiety. None of the compounds are more active in comparison to the reference drug, cisplatin, but the values of some compounds are close to the cisplatin values.

As a conclusion, the compound 30, 3-{4-chloro-2-[(4-chlorophenyl)methyl-361 thio]-5-methylbenzenesulfonyl}-2-[3-(4-hydroxyphenyl)-1-phenylprop-2-enylidene-362 amino]guanidine was proposed as a new lead structure in this work for the potential treatment of the colon cancer.

My suggestions on the present manuscript can be seen in the following parts below:

Grammatical suggestions:

23, 27, 167, 172, 364: “fold” could be used instead of “times”.

28: …than those of HaCaT cells.

40: no need “…one of…”, because among is still present in the beginning of the sentence.

40: “one” could be better after “the most common”.

44: “characterize” should be passive voice due to the “by”.

46: need for “among others”??

56: “Some” instead of “Selected”? (two past participle, selected known).

65: The long description of the PTSA abbreviation should be given at least once in the text or on the Scheme 2.

79: missing paranthesis in “ - C=CH) ”.

81: If the protons are written by using hypen (H-3), then this should be applied for all protons.

Scientific Suggestions:

150-151: It is obvious that R2 substituent has strong influence on the cytotoxic activity. However, the interpretation “R1 group appears to be of minor importance” should not be included as a precise conclusion. Because, we do not have enough number of constant R2 moiety in order to make comparisons. Yes, it seems like a minor importance but not precise.

177: It is clear that the new compounds target the cancer cells more than the healthy ones, but a reference value is also needed for Table 2? e.g. selectivity data for cisplatin could also be better?? Because, the IC50 values were compared to the cisplatin data.

Supplementary Material: Mass spectrometry data-spectra and the 13C NMR spectra-data of the synthesized compunds may be also included in the supporting information part. How did the authors decide to give some spectra of only some compounds, based on which criteria?

Author Response

Thank you for your insightful comments and suggestions. We have corrected the manuscript according to your grammar suggestions.

Regarding your scientific suggestions, we have made the following corrections:

  • 150-151: It is obvious that R2 substituent has strong influence on the cytotoxic activity. However, the interpretation “R1 group appears to be of minor importance” should not be included as a precise conclusion. Because, we do not have enough number of constant R2 moiety in order to make comparisons. Yes, it seems like a minor importance but not precise.

We have corrected this as follow:

The results clearly indicated that cytotoxic activity is observed only for compounds with the hydroxyl group as R2 substituent.

  • 177: It is clear that the new compounds target the cancer cells more than the healthy ones, but a reference value is also needed for Table 2? e.g. selectivity data for cisplatin could also be better?? Because, the IC50 values were compared to the cisplatin data.

We have provided data for ciplatin cytotoxicity against HaCaT cells. As you can find, ciplatin is significantly more toxic against healthy cells than descibed derivatives.

  • Supplementary Material: Mass spectrometry data-spectra and the 13C NMR spectra-data of the synthesized compunds may be also included in the supporting information part. How did the authors decide to give some spectra of only some compounds, based on which criteria?

We have completed missing data as you reccomended. In the first version of manuscript we gave some examples but in the revised version you can find spectra for all new compounds.

Reviewer 2 Report

The topic of the work is very interesting, and the results achieved by the authors are of great importance considering that the tested compounds showed antitumor activity with IC50 values below 15μM. Also, selectivity, which is one of the crucial indicators of good antitumor potential, was very good. However, in addition, revision of the work is needed. Below are the comments. If the authors revise the Manuscript considering the comments below, then I suggest that the paper can be accepted for publication in the International Journal of Molecular Sciences.

The authors should report 1H and 13C NMR spectra for all compounds.

All newly synthesized compounds should be fully characterized, which implies the mass spectrometry.

The authors should explain why they tested the molecules on these specific cell lines.

My suggestion is, if the authors are able, to examine interactions with transport proteins such as HSA or BSA, which would significantly improve the quality of this research work.

Some minor objections are that revision of the English language should be done.

Author Response

Thank you for your insightful comments and suggestions. Regarding your review, we have made the following corrections:

  • The authors should report 1H and 13C NMR spectra for all compounds. All newly synthesized compounds should be fully characterized, which implies the mass spectrometry.

We have completed missing data as you reccomended.

  • The authors should explain why they tested the molecules on these specific cell lines.

The studied cell lines displayed a remarkable sensitivity towards derivatives with
2-mercaptobenzenesulfonamide scaffold as it has been shown in our previous studies. Furthermore, the epidemiological data indicate that cervical, breast and colon cancers are one of the most common cases. Due to WHO data, colorectal cancer is the third most common cancer in the world. Breast cancer is the most common cancer in women worldwide and represents about 12% of all new cancer cases and 25% of all cancers in women. Cervical cancer, in turn, is the fourth most common cancer in women.

  • My suggestion is, if the authors are able, to examine interactions with transport proteins such as HSA or BSA, which would significantly improve the quality of this research work.

We are grateful for this valuable advice. Unfortunately, we are not able to perform these studies at this moment. We will consider to perform similar research for our next projects.

Round 2

Reviewer 2 Report

Dear Editor,

I would like to thank you for inviting me to review the paper. This is a revised version of a submitted work. It seems that the authors have tried to improve the quality of the manuscript according to the comments. They have also added some important experiments to support the results and conclusions of the work. Now I think the work is appropriate and can be considered for publication in IJMS.

Author Response

We would like to thank the Reviewer for the valuable comments and appreciation of our work.